# Comparison of a VR Stylus with a Controller, Hand Tracking and a Mouse for Object Manipulation and Medical Marking Tasks in Virtual Reality

**ABSTRACT**

For medical surgery planning, virtual reality (VR) provides a new kind of user experience, where 3D images of the operation area can be utilized. Using VR, it is possible to view the 3D models in a more realistic 3D environment, which would reduce the perception problems and increase spatial understanding. In the present experiment, We compared a mouse, hand tracking, and a combination of a VR stylus and a VR controller as interaction methods in VR. The purpose was to study the viability of the methods for tasks conducted in medical surgery planning in VR. The tasks required interaction with 3D objects and high marking accuracy. The stylus and controller combination was the most preferred interaction method. In subjective results, it was considered as the most appropriate, while in objective results, the mouse interaction method was the most accurate.

**Index Terms:** Human-centered computing—Human computer interaction (HCI)—Interaction devices—Pointing devices; Human-centered computing—Human computer interaction (HCI)—Empirical studies in HCI; Human-centered computing—Human computer interaction (HCI)—Interaction paradigms—Virtual reality

## 1 INTRODUCTION

Virtual reality makes it possible to create computer-generated environments that replace the real world. For example, the user can interact with virtual object models more flexibly using various interaction methods than with real objects in the real environment. VR has become a standard technology in research, but it has not been fully exploited in professional use even if its potential has been demonstrated.

In the field of medicine, x-ray imaging is routinely used to diagnose diseases and anatomical changes as well as for scientific surveys [31]. In many cases 2D medical images are satisfactory, but they can be complemented with 3D images for more complex operations where detailed understanding of the 3D structures is needed.

When planning surgeries, medical doctors, surgeons, and radiologists study 3D images. Viewing the 3D images in 2D displays can present issues to control object position, orientation, and scaling. Using VR devices, like head mounted displays (HMD), 3D images can be more easily perceived when viewed and interacted with in a 3D environment than with a 2D display. For the medical professionals to be able to do the same tasks in VR as they do in 2D, the interaction methods need to be studied properly. The interaction method needs to be accurate, reasonable, and suitable for the medical tasks. Because we talk about medical work, the accuracy is crucial to avoid as many mistakes as possible. König *et al.* [21] studied an adaptive pointing for the accuracy problems caused by hand tremor when pointing distant objects. The used interaction method needs also to be natural so that the doctors would use it in their daily work and so that they still can focus on their primary tasks without paying too much attention to the interaction method. One typical task for the doctors is marking anatomical structures and areas on the surface of the 3D model. The marked points create the operative area, or they can be used for training.

For 2D content, a mouse is one of the best options for interaction due to its capability to point at small targets with high accuracy and the fact that many users are already very experienced with this device [27]. Mouse cursor can be used for 3D pointing with ray-casting [34] which allows pointing of the distant objects as well. The familiarity and accuracy make the mouse a worthy input method in VR, even though it is not a 3D input device. In addition, controllers have been identified as an accurate interaction method [13, 17] and they are typically used in VR environments [22]. Controllers enable direct manipulation, and the reach of distant objects is different than with the mouse with ray-casting. Other devices, like styluses have been studied in pointing tasks previously [27, 40]. Therefore we aimed to investigate performance of a stylus together with a controller in selected tasks.

The cameras and sensors on HMD devices also allow hand tracking without hand-held input devices. Pointing at objects with a finger is natural way of acting for humans, so hand interaction can be expected to be received well. Hand interaction was selected as one of the conditions based on interviews of medical professionals and their expectations for the supporting technology.

We decided to use a marking task to assess the three interaction conditions. The conditions were a standard mouse, bare hands, and a handheld controller with VR stylus. All methods were used in a VR environment to minimise additional variation between methods and to focus the comparison on interaction techniques. The use of the HMD also allowed the participants to easily study the target from different directions by moving their head. In the medical marking task the doctor will observe the anatomical structures by turning and moving the 3D object and at the same time looking for the best location for the mark. The time spent for the manipulation is not easily separated from the time spent in the final marking. The doctor decides during the manipulation from which angle and how the marking will be done, which will affect the marking time. This made application of Fitts' law [11] not possible in our study, as it requires that a participant cannot influence target locations.

We had 12 participants who were asked to do simplified medical surgery marking tasks. To study the accuracy of the interaction methods, we created an experiment where in the 3D model there was a predefined target that was marked (pointed+selected). In the real medical case, the doctor would define the target, but then the accuracy cannot be easily measured. This study focused mainly on subjective evaluations of interaction methods, but also included objective measurements.

The paper is organized as follows: First, we go through background of object manipulation and marking, interaction methods in 3D environment, and jaw osteotomy surgery planning (Section 2). Then, we introduce the compared interaction methods and used measurements (Section 3), as well as go through the experiment (Section 4) including apparatus, participants, and study task. In the end the results are presented (Section 5) and discussed (Section 6).

## 2 BACKGROUND

### 2.1 Object manipulation and marking

Object manipulation, i.e. rotating and translating the object in 3D space, and object marking, i.e. putting a small mark on the surface

of an object, have been used as separate task when different VR interaction methods have been studied. Sun *et al.* [32] had 3D positioning task that involved object manipulation. When a mouse and a controller were compared for precise 3D positioning the mouse was found as the more precise input device. Object marking has been studied without manipulation in [27]. Argelaguet and Andujar [1] studied 3D object selection techniques in VR and Dang *et al.* [9] have studied 3D pointing techniques. As there are no clear standard techniques for 3D object selection nor 3D pointing technique, Argelaguet and Andujar and Dang *et al.* attempt to bring practices in studying new techniques in 3D UIs.

In earlier work using bimanual techniques, Balakrishnan and Kurtenbach [5] presented a study where dominant and non-dominant hand had their own tasks in a virtual 3D scene. The bimanual technique was found as faster and preferable. People typically use their both hands to cooperatively perform the most skilled tasks [5, 12] where the dominant hand is used for the more accurate functions, and the non-dominant hand sets the context such as holding a canvas when dominant hand is used to draw. The result is optimal when bimanual techniques are designed by utilizing the strengths of both dominant and non-dominant hands.

## 2.2 Input devices for object manipulation and marking

### 2.2.1 Mouse

A mouse is a common, familiar, and accurate device for 2D content to point at small targets with high accuracy [27]. The mouse is also a common device to do medical surgery planning [22]. Many studies have used a mouse cursor for 3D pointing with ray-casting [6, 8, 22, 27, 34]. Ray-casting technique is easily understood, and it is a solution for reaching objects at a distance [25].

Compared to other interaction methods in VR, the issue of the discrepancy between the 2D mouse and a 3D environment has been reported [1], and manipulation in 3D requires a way to switch between dimensions [4]. Balakrishnan *et al.* presented Rocking'Mouse to select in 3D environment while avoiding a hand fatigue. Kim and Choi [20] mentioned that the discrepancy creates a low user immersion. In addition, use of a mouse usually forces the user to sit down next to a table instead of standing. The user can rest their arms on the table while interacting with the mouse which decrease hand fatigue. Johnson *et al.* [18] stated that fatigue with mouse interaction will appear only after 3 hours.

Bachmann *et al.* [3] found that Leap Motion controller has a higher error rate and higher movement time than the mouse. Kim and Choi [20] showed in their study that 2D mouse have high performance in working time, accuracy, ease of learning, and ease of use in VR. Both Bachmann *et al.* and Kim and Choi found the mouse to be accurate but on the other hand Li *et al.* [22] pointed that with difficult marking tasks small displacement of a physical mouse would lead to a large displacement on the 3D model in the 3D environment.

### 2.2.2 Hands

Hand interaction is a common VR interaction method. Voigt-Antons *et al.* [39] compared free hand interaction and controller interaction with different visualizations. Huang *et al.* [17] compared different interaction combinations between free hands and controllers. Both found that hand interaction has lower precision than the controller interaction. With alternative solutions like a Leap Motion controller [28, 41] or using wearable gloves [42] the hand interaction can be done more accurately. Physical hand movements create a natural and realistic experience of interaction [10, 17], and therefore hand interaction is still an area of interest.

### 2.2.3 Controllers

Controllers are the leading control inputs for VR [17]. When using controllers as the interaction method, marking, and selecting

are usually made with some of the triggers or buttons on the controller. Handheld controllers are described as stable and accurate devices [13, 17]. However, holding extra devices in hands may become inconvenient, if the hands are needed for other tasks between different actions. When interacting with hands or controllers in VR, the fatigue in arms is one of the main issues [1, 15]. Upholding arms and carrying the devices also increase the arm fatigue.

### 2.2.4 VR stylus

A VR stylus is penlike handheld device that is used in VR environment as a controller. The physical appearance of Logitech VR Ink stylus [23] is close to a regular pen except it has buttons which enables different interaction e.g., selecting, in VR. Batmaz *et al.* [7] have studied Logitech VR Ink stylus for a selection method in virtual reality. They found that using a precision grip there is no statistical differences on the marking if the distance of the target is changing. Wacker *et al.* [40] presented as one of their design VR stylus for mid-air pointing and selection happened pressing a button. For object selection, the users preferred a 3D pen over a controller in VR [27].

## 2.3 Jaw osteotomy surgery planning

Cone Beam Computed Tomography (CBCT) is a medical imaging technique that produce 3D images that can be used in virtual surgery planning. Compared to previous techniques that were used in medical surgery planning like cast models, virtual planning with CBCT images has extra costs and time requirements [14]. However, the technique offers several advantages for planning accuracy and reliability [31]. CBCT images can be used as 3D objects in VR for surgery planning with excellent match to real objects [14]. Ayoub and Pulijala [2] reviewed different studies about virtual and augmented reality applications in oral and maxillofacial surgeries.

In virtual surgery planning, the procedures for surgery are implemented and planned beforehand. The real surgery is done based on the virtual plan. Common tasks in dental planning are specifying the location of impacted teeth, preventing nerve injuries, or preparing guiding flanges [31]. In VR this can be done by marking critical areas or drawing cutting lines on to the models. Virtual planning can be used in student education as well, where the procedures can be realistically practiced. Reymus *et al.* [29] found that students understood the mouth anatomy better after studying 3D models in VR environment than from regular 2D image. The objects can be closer, bigger, and they can move in depth direction in 3D environment compared to 2D environment [19].

Tasks, like understanding the 3D object and marking critical areas on it need to be done in medical surgery planning. However, working with 3D objects in 2D environment makes the task more difficult. Hinckley *et al.* [16] studied issues for developing effective free-space 3D user interfaces. Appropriate interaction and marking methods help to understand 3D objects and perform the required tasks in VR. In this study, we evaluated three methods for VR object manipulation and marking and examined the performances in simplified medical surgery planning tasks.

## 3 METHOD

### 3.1 Mouse

In the first interaction method *a regular mouse* was used inside a VR environment (Figure 1). In VR environment there was a visualized mouse model that the participant was able to move by manipulating the physical mouse and to control the direction of a ray starting from the model. The ray was always visible in Mouse interaction.

Mouse was used one-handed when the other two methods were two-handed. Mouse was used to perform two functions, manipulation and marking, while these functions had been separated in other methods into different hands. In addition, Mouse used ray-casting, ray from the mouse, while the two other methods did not use it. The other methods used direct mid-air object manipulation.

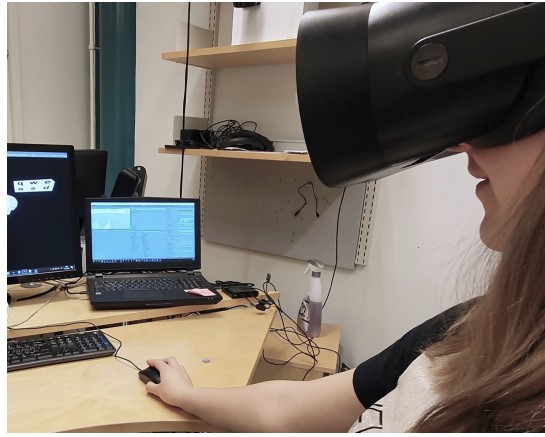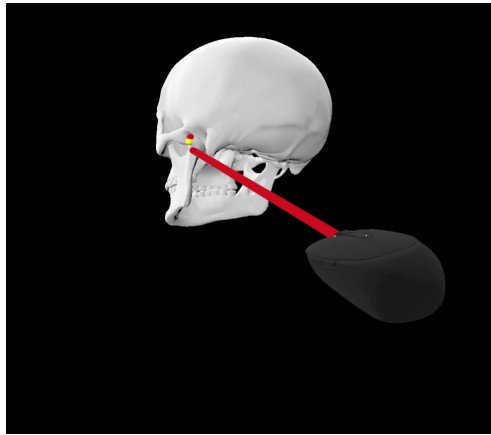

Figure 1: Mouse interaction method outside VR (left). Mouse marking method inside VR and the study task (right).

The participant could rotate the object in 3 dimensions using the mouse movements with right click. For the 3D translations the participant used the scroll button. Using the scroll wheel, the user can zoom in and out (translate in Z) and when the user presses the scroll button and moves the mouse, the user can translate up-down and sideways (translate in X and Y). Markings were made by pointing the target with the ray and pressing the left button.

For the real-world mouse to be visible inside VR, pass through is not really required even though the mouse was visible in our study. After wearing the headset, the user could see the virtual mouse that is positioned to where the physical mouse is located to be able to find and reach the device. When the user moved the physical mouse sideways, the movement was converted to a horizontal rotation of the beam from the virtual mouse, and when the mouse was moved back and forth, the movement was converted to a vertical rotation of the beam. This way the user can cover a large space similar to using mouse in 2D displays. To improve ergonomics, the user could configure the desk and chair for their comfort.

### 3.2 Hands

As the second interaction method, the participant used bare *hands*. The left hand was for object manipulation and the right hand for object marking. The participant could pick up the 3D object by a pinch gesture with their left hand, to rotate and move the object. Marking was done with a virtual pen. In the VR environment the participant had the virtual pen attached to their right palm, near to the index finger (Figure 2 right). As the palm was moved the pen moved accordingly. When the virtual pen tip was close to the target, the tip changed its color to green to show that the pen was touching the surface of the object. The mark was put on the surface by bending the index finger and pressing the pen's virtual button. The participant had to keep their palm steady when pressing the button to prevent the pen from moving.

### 3.3 Controller and VR stylus

The third interaction method was based on having *a controller* on participant's left hand for the object manipulation and *a VR stylus* on the right hand for the marking (Figure 3). The participant grabbed the 3D object with hand grab gesture around the controller to rotate and move the object. The markings were made with the physical VR stylus. The VR stylus was visualized in VR as was the mouse, so the participant knew where the device was located. The participant pointed the target with the stylus and pressed its physical button to make the mark. The act of press was identical to the virtual pen press in Hands method. There was a passive haptic feedback when touching the physical VR stylus, which did not happen with the virtual pen.

There have been some supporting results for using mouse in VR [3, 20, 22, 25] but 2D mouse is not fully compatible with the 3D environment [20]. We studied the ray method with Mouse to compare it against Hands and Controller+Stylus for 3D object marking. We also compared Hands without any devices to a method with a device in one or two hands. The marking gesture was designed to be similar in Hands and Controller+Stylus methods to be able to compare the effect of the devices.

### 3.4 Measurements and the pilot study

The participant was asked to make a marking as close to the target location as possible. We used Euclidean distance to measure the distance between the target and the participant's marking. The task completion times were measured. The participant was able to remark the target if s/he was dissatisfied with the current marking. We counted how many remarkings were made to see if any of the interaction methods required more remarking than the other methods. We measured accuracy in these two ways, as a distance from the target and as the number of dissatisfied markings.

A satisfaction questionnaire was filled after each interaction method trial. There were a question and seven satisfaction statements that were evaluated on a Likert scale from 1 (strongly disagree) to 5 (strongly agree). The statements were grouped so that the question and the first statement were about the overall feeling and the rest of the statements were about object manipulation and marking separately. The statements were:

- Would you think to use this method daily?

- Your hands are NOT tired.

- It was natural to perform the given tasks with this interaction method.

- It was easy to handle the 3D objects with this interaction method.

- The interaction method was accurate.

- The marking method was natural.

- It was easy to make the marking with this marking method.

- The marking method was accurate.

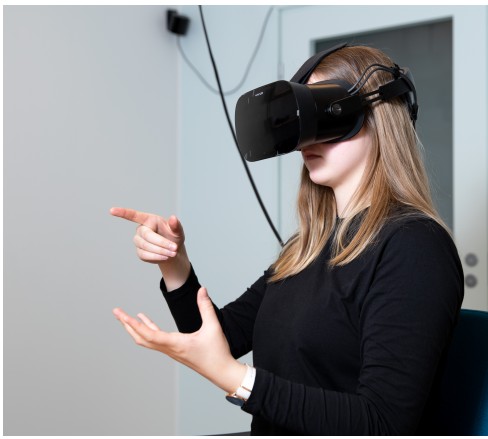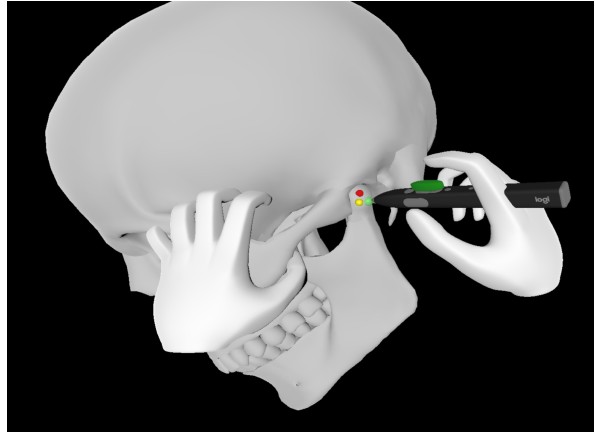

Figure 2: Hands interaction method outside VR (left). Hands marking method inside VR and the study task (right).

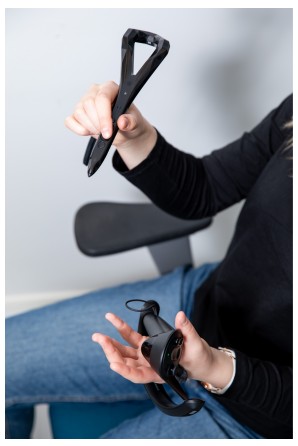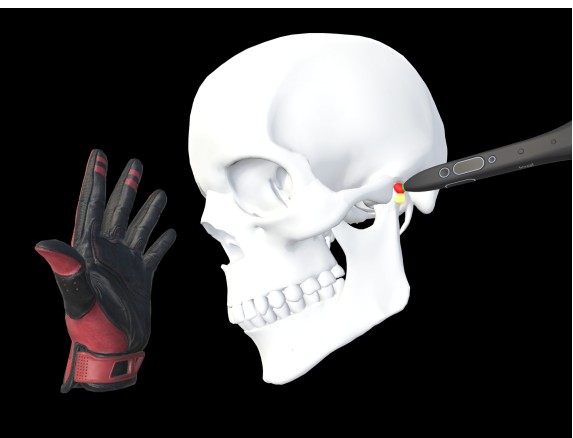

Figure 3: Controller interaction method outside VR (left). Stylus marking method inside VR and the study task (right).

The statements were designed to measure fatigue, naturalness, and accuracy as they have been measured in earlier studies [1, 10, 17] as well. Accuracy was measured also from data to see if the objective and subjective results are consistent. With these statements, it was possible to measure the easiness and ability to use the method daily unlike from objective data.

In the questionnaire there were also open-ended questions about positive and negative aspects of the interaction method. In the end the participant was asked to rank the interaction methods in order from the most liked to the least liked.

A pilot study was arranged to ensure that tasks and the study procedure were feasible. Based on the findings in the pilot study, we modified the introduction to be more specific and added a mention about the measured features. We also added the ability to rotate the 3D object even after the mouse ray moved out of the object. The speed of the mouse ray in VR environment was increased to better match the movements of the real mouse.

## 3.5 Statistical measures

We used two different statistical tests to analyze possible statistically significant differences between different parameter sets. For objective data (completion times, number of markings, and accuracy) we used the paired t-test. For data from evaluation questionnaires (fatigue, daily use, naturalness, easiness, and subjective accuracy) we first used Friedman test to see if any statistically significant differences appeared, and then we used the Wilcoxon signed rank test

as it does not assume the numbers to be in ratio scale or to have normal distribution.

The study software saved the resolution of time in milliseconds and the resolution of distances in meters. To clarify the analysis, we transferred these to seconds and millimeters.

## 4 EXPERIMENT

### 4.1 Participants

We recruited 12 participants for the study. The number of participants was decided based on a power analysis for paired t-test and the Wilcoxon signed rank test, assuming large effect size, a power level of 0.8 and an alpha level of 0.05. The post hoc calculated effect sizes (Cohen's d or R value, for paired t-test or Wilcoxon signed rank test, respectively) are reported together with the p-values in Results Section 5 for comparison to the assumption of large effect size. 10 of the participants were university students and two were full time employees, on the field not related to medicine or dentistry. The ages varied from 21 to 30 years, mean age was 25 years. There were 6 female participants and 6 male participants. Earlier VR experience was asked on a scale from 0 to 5, and the mean was 1.75. Two participants did not have any earlier experience. One participant was left-handed but was used to use the mouse with the right hand. Other participants were right-handed.

### 4.2 Apparatus

#### 4.2.1 Software, hardware, and hand tracking

The experiment software was built using the Unity software [35]. With all methods we used Varjo VR2 Pro headset [37], which has an integrated vision based hand tracking system that was used for Hands interaction. Hands were tracked by Ultraleap Stereo IR 170 sensor mounted on a Varjo VR2 Pro. For the Controller+Stylus, we used Valve Index Controller [36] together with Logitech VR Ink stylus [23]. These were tracked by SteamVR 2.0 base stations [38] around the experiment area.

#### 4.2.2 Object manipulation and object marking

The study task combined two phases: object manipulation phase where the object was rotated and translated in 3D space and object marking phase where a small mark was put on the surface of an object. In object manipulation phase the participant either selected the 3D object by mouse ray or pinched or grabbed the 3D object with hand gesture. The 3D objects did not have any physics and laid in mid-air. By rotating and translating the object the participant can view the object from different angles. The participant can also use head moves to change their point-of-view.

Instead of only pointing the target, the marking needs to be confirmed. This allows us to measure the marking accuracy and if the user understood the 3D target's location related to the pointing device. The participant could either release the 3D object in mid-air or hold it in their hand when Hands or Controller+Stylus was used in object marking task. The marking was done either pointing by mouse ray and clicking with Left click, touching the target with virtual pen, and marked with a hand gesture, or touching and marking with the VR stylus.

### 4.3 Procedure

First, the participant was introduced to the study, s/he was asked to read and sign a consent form, and fill in a background information form. For all conditions the facilitator would demonstrate him/herself the system functions and the controls. Each participant had an opportunity to practice before every condition. The practice task was to move and rotate a cube having several target spheres, and to mark those targets as many times as needed to get to know both the interaction and the marking methods. After the participant felt confident with the used method, s/he was asked to press the Done button, and the real study task appeared.

The participant was asked to find and mark a hidden target on the surface of each 3D object model. The target was visible all the time whereas the participant's marking was created by the participant. When the target was found it was first pointed and then marked. The aim was to place the participant's mark (a yellow sphere) inside the target sphere (red) (see Figures 1 right, 2 right, and 3 right). Each 3D object had one target on it and the task was repeated five times per each condition. The order of 3D objects was the same to all participants: lower jaw, heart, skull, tooth, and skull. The order of interaction methods was counter-balanced between the participants using balanced Latin Squares. This was done to compensate possible learning effects. The target locations on the 3D object were predefined and presented in the same order for the participants.

The used task needed both object manipulation (rotating and translating) and marking (pointing and selecting). By combining the manipulation and marking tasks together, we wanted to create a task that simulates a task that medical professionals would do during virtual surgery planning. Both the object manipulation and marking are needed by the medical professionals. The marking is relevant when selecting specific locations and areas of a 3D model and it requires accuracy to make the marks in relevant locations. This medical marking task does not differ from regular marking tasks in other contexts as such, but the accuracy requirements are higher. By

manipulating the 3D model, the professional has an option to look at the pointed area from different angles to verify its specific location in 3D environment.

A satisfaction questionnaire was filled after each interaction method trial, and after all three trials, a questionnaire was used to rank the conditions.

## 5 RESULTS

In this section, we report the findings of the study. First, we present the objective results from data collected during the experiment, and then the subjective results from the questionnaires.

### 5.1 Objective results

The task completion times (Figure 4, top left) include both object manipulation and marking, and it had some variation, but the distributions of median values for each interaction method were similar and there were no significant differences. The completion time varied slightly depending on how much VR experience the participant had before, but there were no statistically significant differences.

The number of markings done before the task completion varied between the interaction methods (Figure 4, top right). The median values for Mouse, Hands, and Controller+Stylus conditions were 6.5, 12, and 7 markings, respectively. However, there were no statistically significant differences. Some participants did many markings in a fast pace (2-3 markings per second) leading to a high number of total markings.

There were some clear differences in final marking accuracy between the interaction methods (Figure 4, bottom). The median values for Mouse, Hands, and Controller+Stylus methods were 3.2, 5.9, and 4.2 millimeters, respectively. The variability between participants was highest with Hands method. We found statistically significant difference between the Mouse and Hands methods (p-value 0.004, Cohen's d 1.178[1]) using a paired t-test and Bonferroni corrected p-value limit 0.017 (= 0.05 / 3). There were no statistically significant differences between the Mouse and Controller+Stylus methods or Hands and Controller+Stylus methods.

### 5.2 Subjective data

Friedman tests showed statistically significant differences in daily use (p-value 0.002), interaction naturalness (p-value 0.000), interaction easiness (p-value 0.001), interaction accuracy (p-value 0.007), marking easiness (p-value 0.039), and ranking (p-value 0.000). There were no significant differences in marking naturalness or marking accuracy. In evaluations for tiredness there were no significant differences (Figure 5, left). Most participants did not feel tired using any of the methods, but the experiment was rather short.

In pairwise tests of everyday use using Wilcoxon signed rank test we found significant differences (Figure 5, right). We found statistically significant differences between the Mouse and Controller+Stylus methods (p-value 0.015, R 0.773[2]) and between Hands and Controller+Stylus methods (p-value 0.003, R 1.000). There were no statistically significant differences between the Hands and Mouse methods or Hands and Controller+Stylus methods.

We asked the participants to evaluate both object manipulation and marking separately. In object manipulation evaluation, there were statistically significant differences in naturalness between Controller+Stylus and Mouse (p-value 0.003, R 1.000) and Controller+Stylus and Hands (p-value 0.009, R 0.879). There was no statistically significant difference between Mouse and Hands. In object manipulation easiness Controller+Stylus had statistically significant difference between Mouse and Hands (p-values 0.003, R 1.000 in both methods), see Figure 6. There were no no statistically significant differences between Mouse and Controller+Stylus

---

[1]Cohen's d $\geq$ 0.8 is considered a large effect size

[2]R value $\geq$ 0.5 is considered a large effect size

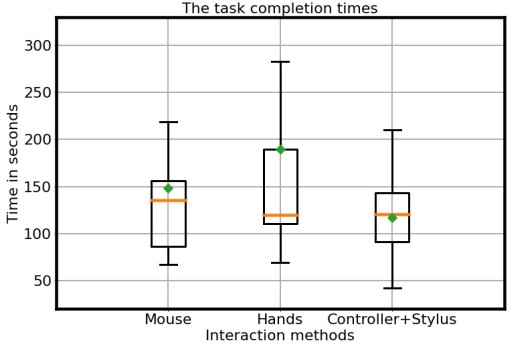

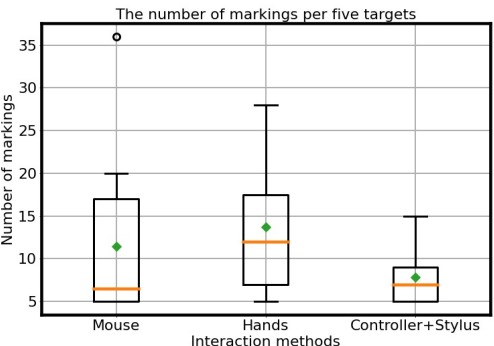

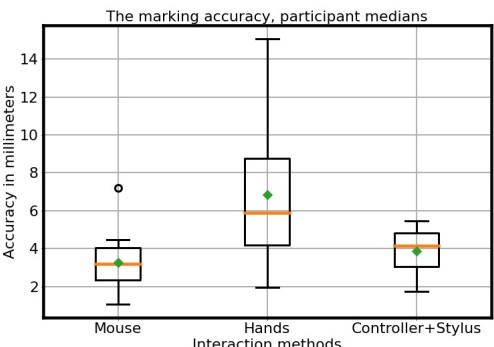

Figure 4: The task completion times for different conditions (top left). The median values for each participant are rather similar between the methods. There were two outlier values (by the same participant, for Mouse and Hands conditions) that are removed from the visualization. The number of markings per five targets (top right). There were some differences between the interaction methods (the median value for Hands was higher than for the other methods), but no significant differences. The marking accuracy (bottom). There were some clear differences between the interaction methods in the final marking accuracy.

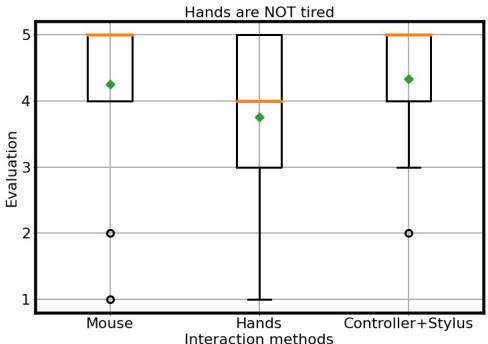

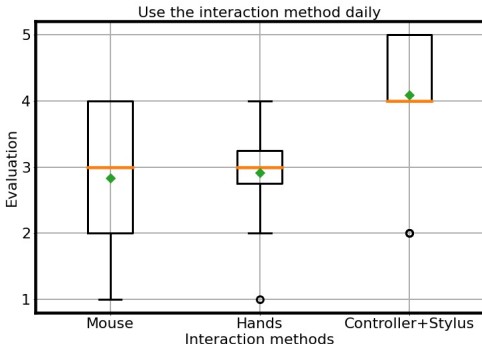

Figure 5: The evaluation of fatigue (left). None of the methods were found to be particularly tiring. The evaluation of possible daily use (right). Controller+Stylus was significantly more usable for daily use than the other methods.

or Hands and Controller+Stylus. In manipulation accuracy evaluation we found statistically significant difference between Controller+Stylus method and Hands method (p-value 0.003, R 1.000). There were no no statistically significant differences between Mouse and Controller+Stylus or Hands and Mouse. In the object marking evaluation (Figure 7), the only significant difference was measured between Controller+Stylus method and Mouse method in easiness (p-value 0.009, R 1.000). There were no no statistically significant differences between Hands and Controller+Stylus or Hands and Mouse.

Multiple participants commented that the controller interaction

felt stable and that it was easy to move and rotate the 3D model with the controller. The participants also commented that holding a physical device in hand so that its weight could be felt increased the feel of naturalness. Not all comments agreed, when one participant felt VR stylus as accurate while another participant said it felt clumsy.

When asked 11 out of 12 participants ranked Controller+Stylus the most liked method. The distribution of ranking values is shown in Table 1. The ranking values of Controller+Stylus method were statistically significantly different to Mouse (p-value 0.008, R 0.885) and Hands (p-value 0.003, R 1.000). There was no statistically significant difference between Mouse and Hands.

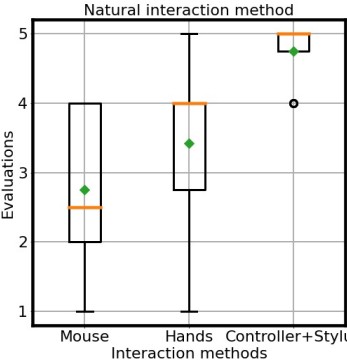 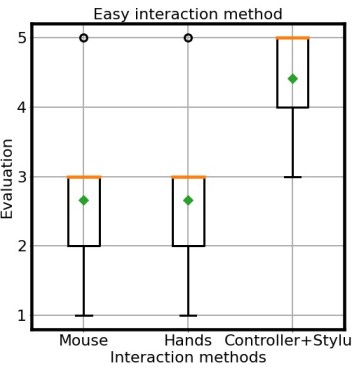 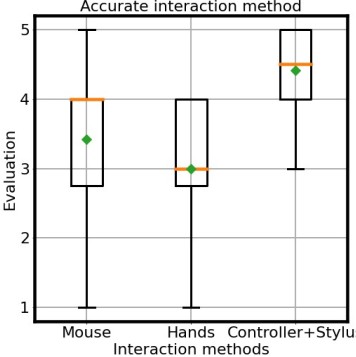

Figure 6: The evaluation of interaction method naturalness (left), easiness (middle), and accuracy (right). Controller+Stylus was the most liked method in these features.

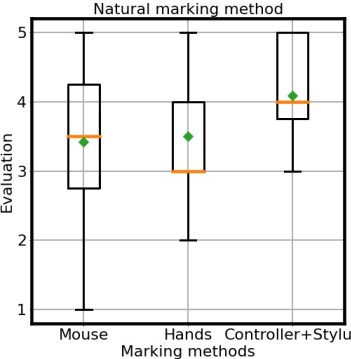 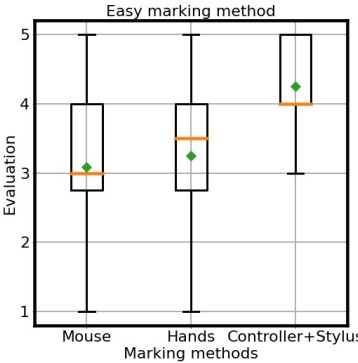 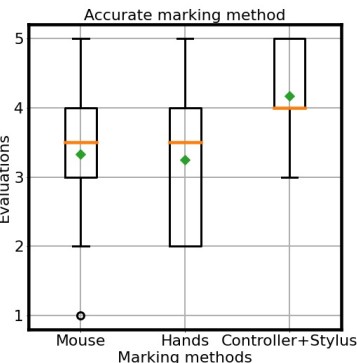

Figure 7: The evaluation of marking method naturalness (left), easiness (middle), and accuracy (right). Median values in these features are rather similar, and significant difference was found only in marking easiness.

Table 1: The number of mentions of different rankings of the interaction methods when asked for the most liked ($1^{st}$), the second most liked ($2^{nd}$), and the least liked ($3^{rd}$) method.

| Condition | Ranking | | |
|---|---|---|---|
| | $1^{st}$ | $2^{nd}$ | $3^{rd}$ |
| Mouse | 1 | 7 | 4 |
| Hands | 0 | 4 | 8 |
| Controller+Stylus | 11 | 1 | 0 |

## 6 DISCUSSION

In this study, we were looking for the most feasible interaction method in VR for object manipulation and marking in a medical context. Controller+Stylus method was overall the most suitable for a task that requires both object manipulation and marking. Controller+Stylus method was the most liked in all subjective features, while Mouse and Hands conditions were evaluated very similarly. The smallest number of markings were done with Controller+Stylus, but no significant differences were found. There were statistically significant differences between the methods in daily use, interaction naturalness, and easiness. Controller+Stylus was statistically significantly more accurate in object manipulation than Hands (p-value 0.003), and easier to use than Mouse (p-value 0.003). Without earlier experience with the VR stylus, the participants had difficulties in finding the correct button when marking with the stylus. The physi-

cal stylus device cannot be seen when wearing the VR headset and the button could not be felt clearly. Even though Controller+Stylus combination was evaluated as natural and the most liked method in this study, the hand-held devices may feel inconvenient [17]. In our study, some participants liked the physical feel of devices. However, our result was based on the subjective opinions of participants, and that might change depending on the use case or devices.

There are many possible reasons for the low hand tracking accuracy. Hand inaccuracy can be seen in the large number of markings and large distribution in task completion times with Hands as the participants were not satisfied with their first marking. Hands were the only method where only one participant succeeded with a minimum of 5 markings, when by other methods, several participants succeeded in the task with 5 markings. One explanatory factor can be the lack of hand tracking fidelity that also has been noticed in other studies [17, 42]. In addition, inaccuracy in human motor system leads to the inaccuracy of hands [15]. The vision based hand tracking system that uses camera on HMD does not recognize the hand gesture well enough and as a result, the participant must repeat the same gesture or movement multiple times to succeed. This extra work also increases the fatigue in Hands. Even though the fatigue were low with all interaction methods, this study did not measure the fatigue of long-term activity. These are clear indications that Hands interaction needs further development before it can be used in tasks that needs high marking accuracy. Several earlier studies have reported the hands inaccuracy compared to controllers [15, 17, 42].

Passive haptics were available with Mouse and when marking

with VR stylus. With Hands there was only visual feedback. The lack of any haptic feedback might have affected the marking accuracy as well because the accuracy was much better with the physical stylus. Li *et al.* [22] found that with the low marking difficulty, the mouse with 2D display was faster than the kinesthetic force feedback device in VR. For high marking difficulty the other VR interface that used a VR controller with vibrotactile feedback was better than the 2D interface. They found that mouse in 2D display has fast pointing capability but in our study, the task completion times did not vary between Mouse and the other methods. Li *et al.* described the fact that manipulating viewing angle is more flexible when wearing HMD than with a mouse in 2D display. In VR interfaces the participant can rotate the 3D object while changing the viewing angle by moving their head. In our study, all methods used HMD, so change of viewing angle was as equally flexible.

Mouse was statistically significantly more accurate marking method than Hands. Mouse was not affected by some of the issues that were noticed with Hands or Controller+Stylus. With Mouse it was not felt problematic that the device cannot be seen during the use. There were no sensor fidelity issues with Mouse, and Mouse was a familiar device to all participants. Only the ray that replaced the cursor was an unfamiliar feature and caused some problems. We found that the ray worked well with simple 3D models but there were a lot of difficulties with complex models where the viewing angle needed to be exactly right to reach the target. If any part of the 3D model blocked the ray, the target could not be marked. When the target was easy to mark the accuracy using Mouse was high. It can be stated that Mouse was an accurate method in VR but for all other measured properties of Controller+Stylus were measured to be better.

Both the target and the marking were spheres in 3D environment. During the study, it was noticed that when a participant made their marking in the same location as the target, the marking sphere disappeared inside the target sphere. This caused uncertainty if the marking was lost or if it was in the center of the target. This may have affected the results when the participants needed to make remarking to be able to see their marking that was not in the center of the target anymore. In future studies the marking sphere should be designed bigger size than the target and transparent so that the participant can be sure about the location of both spheres.

Our focus was in comparing three different interaction and marking methods and their suitability for the medical marking task. To simplify the experimental setup, the experiment was conducted with simplified medical images, which may have led to optimistic results for the viability of the methods. Even then, there were some problems with Mouse interaction method. To further confirm that the results are similar also for more realistic content, a similar study should be conducted in future work with authentic material utilizing, for example, original CBCT images in VR instead of the simplified ones.

Future research may investigate multimodal interaction methods to support even more natural alternatives. Speech is the primary mode for human communication [30]. Suresh *et al.* [33] used three voice commands to control gestures of a robotic arm in VR. Voice is a well suitable input method in cases where hands and eyes are continuously busy [15]. Pfeuffer *et al.* [26] studied gaze as an interaction method together with hand gestures but found that both hand and gaze tracking still lack tracking fidelity. More work is still needed, as Nukarinen *et al.* [24] stated that human factor issues made the gaze as the least preferred input method in an object selection task in VR.

## 7 CONCLUSION

The 3D medical images can be viewed in VR environments to plan for surgeries with expected results. During the planning process one needs to interact with the 3D models and be able to make markings of high accuracy on them. In this study, we evaluated the feasibility of three different VR interaction methods Mouse, Hands, and Controller+Stylus combination in virtual reality. Based on the results, we can state that Valve Index controller and Logitech VR Ink stylus combination was the most feasible for tasks that require both 3D object manipulation and high marking accuracy in VR. This combination did not have issues with complex 3D models and sensor fidelity was better than with Hands interaction. Statistically significant differences were found between the controller combination and the other methods.

Hand-based interaction was the least feasible for this kind of use according to the collected data. Hands and Mouse methods were evaluated almost equal in the feasibility by participants. With the current technology, free hands usage cannot be proposed for accurate marking tasks. Mouse interaction was more accurate than Controller+Stylus. In detailed tasks Mouse could replace the free hands interaction. The discrepancy between the 2D mouse and the 3D environment needs to be solved before Mouse could be considered a viable interaction method in VR.

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
