# OpenReview forum: "Comparison of a VR Stylus with a Controller, Hand Tracking and a Mouse for Object Manipulation and Medical Marking Tasks in Virtual Reality"
_graphicsinterface.org/Graphics_Interface/2022/Conference — Submitted to GI 2022_

### Official Review · Reviewer_iDY2 · 2022-04-13
**Improved from previous submission, now above the bar**

**Rating:** 6
**Confidence:** 5

**Review:**

(Note: this review considers changes made for the second GI submission cycle; I also reviewed the initial submission).

The current submission addresses several of the problems seen in the original version. The first review identified the following issues:

* related work inadequate: the revision adds several papers to its related work and does a better job of providing a context for the research in terms of interaction techniques. However, Section 2 still focuses primarily on devices rather than the techniques that can arise from those devices, and this could still be improved further. Additional text should also be devoted to comparing the specifics of the two-handed technique in the study with previous techniques. For example, the authors cite Guiard's kinematic chain paper, but do not use it to describe their bimanual technique (and it clearly is relevant).

* experiment description: the revision's description of the experiment is improved, although more detail is still required:
- the motivation for selecting the two non-mouse conditions is still unclear. First, the authors do not provide a real rationale for choosing the controller+stylus device: they only state that these two devices are both used, and and then jump to the conclusion that "Therefore
we aimed to investigate performance of a stylus together with a
controller" - but there is no concrete argument that would lead to this conclusion. A clear justification for why a bimanual technique is worth considering is required (and there is plenty of prior work that could provide the starting point for that justification).
- second, the rationale for the hands technique is missing: the authors state that "Hand interaction was selected as one of the conditions based on interviews of medical professionals and their expectations for the supporting technology" but there are no citations to literature or data regarding these interviews. The authors should either present a summary of their own interviews, or cite a source. (And as a side note, the authors should be clear as to whether they are considering operating-room conditions as a motivation for the hands technique - this would be a justification, but it does not seem likely that surgery planning occurs in the OR).
- the authors have added some explanation for why they chose a marking task rather than a search task, but more information is still needed to clearly explain why the marking task is relevant to surgical planning. If selection of existing marks is an important component of some aspect of the task, this needs to be made clear - otherwise, the connection to surgical planning does not provide any justification for the study tasks.

* study results: the paper provides considerably more information about the study's results, and this is an improvement. However, the following issues should still be addressed:
- the paper states that it will focus on subjective results rather than performance results, but the authors must do more to explicate how their idea of looking for "the most feasible interaction method" is operationalized across different aspects, including both subjective and objective measures. A rationale for focusing on experiential measures rather than performance is required. I note that the strong preference for the combined device is a definite strength, but to answer the authors' research question requires a clearer comparison of the controller+stylus results versus the mouse results.
- in particular, the paper needs to do more to explain why the preference results went in favour of the controller+stylus; this should be done in terms of differences between the techniques (which should be clear from earlier in the paper; see comments about this above)

Overall, the paper is improved from the previous version and there is now an evident contribution; however, more could be done to improve the presentation and justification for the study design, and the discussion of implications of the findings.

---

### Official Review · Reviewer_Djre · 2022-04-13
**This paper presents a comparison of three input devices -- mouse, stylus, and hand gestural -- for selecting points on the surface of a 3D model in VR. The authors found a preference for  the stylus method among their 13 participants. But there was no clear difference between methods in terms of accuracy or speed.**

**Rating:** 5
**Confidence:** 4

**Review:**

The application and input methods are interesting. The results of the
experiment is also interesting but don't help us understand why participants
preferred the stylus.

For example, the stylus was the only device that had haptic feedback. Did participants
notice it and was that a major factor for the preference? What if the mouse
had haptic feedback, e.g. vibration? Alternatively, what if every input mode gave
lots of feedback when the user makes contact with the object, such as playing
a sound or showing marks on the surface of the object. Would this affect
people's preference and speed to perform the task?

The goal of the study is to investigate devices for medical marking in VR. Did
the size of the marker in the experiment reflects the accuracy that a doctor
would need in practice?

Were there differences between the 5 models used for the experiment? Were some
models harder than others to place the mark? Is it true that participants
clicked the same target on the same model 5 times per condition? Why not give
participants different targets? Once the user finds the target and clicks on it
once, isn't it easy to click it again 4 additional times? Could this be why
the accuracy and time are similar for all conditions?  Also, we cannot tell if
rotating is harder with some input methods or whether clicking a target is
harder with some methods than others.

---

### Official Review · Reviewer_13dh · 2022-04-14
**some weaknesses with the study**

**Rating:** 5
**Confidence:** 4

**Review:**

I reviewed an earlier version of this submission.  It has been improved by adding a video, which is very helpful to understand the interaction techniques, and also by citing more related work.  There is some limited justification given in section 1 (paragraphs 4 and 5) for the choice of interaction methods.  I don't find the justification strong, but it is a justification nevertheless.

<p>
There are still two weaknesses with the study that make me not in favor of accepting the work at GI:

<p>
First, if I understand correctly, the users could click (or "mark") as many times as they wanted on or near the target before confirming that they had completed the task, and figure 4 (upper right) shows that often users marked a target 10 or even 15 times before confirming that they had finished the task.  Section 6, 2nd paragraph, confirms that many users did more than 5 marks per target.  Section 6, paragraph 5, then tells us that users were sometimes confused when they had marked inside the target because the marking sphere would disappear inside the target sphere, and this "may have affected the results when the participants needed to [perform] remarking to be able to see their marking that was not in the center of the target anymore".  This kind of confusion should be eliminated during warm-up tasks or practice tasks, and the study did indeed have practice tasks (section 4.3, paragraph 1) but it seems that users had control over when they could proceed to the real tasks.  A better design for the experiment could force users to perform a minimum fixed number of practice tasks.  It is unclear to me how much the results in time and accuracy were affected by this confusion.

<p>
Second, and more significantly, the fact that users often had to mark (click) 5 times or more also tells me that a better user interface would allow the user to mark close to the target, and then do something to make minor corrections, like hitting a button to switch to a lower gain, or hitting a button to switch to relative displacement, or hitting some arrow keys on a keyboard to make very small adjustments to the marked location.  Such a user interface might require much less time to complete the task.

<p>
Additional, minor comments:

<p>
- Section 4.1 states 12 participants were chosen to achieve power of 0.8 assuming a large effect size, but the footnote 1 on the next page states "Cohen's d >= 0.8 is considered a large effect size", and if I use https://statulator.com/SampleSize/ss2PM.html to find sample size for effect size of 0.8, it says we need a sample siz of 16.  On the other hand, for effect size 1.0, we only need a sample size of 11.  So the submission should clarify what is meant in section 4.1 by "large effect size".

<p>
- p values are not normally reported as "p-value 0.002", but rather as "p < 0.002" or "p = 0.002".  A p value of "0.000" should not be reported, as p cannot be zero, it can only be bound above (p < ...).

<p>
- Figure 4, top left: are these the times for each user and for each task and for each method, or are these times the total time over the 5 tasks for each user and for each method?  Should we divide the times by 5 to obtain the time for each user and each task and each method?  What about the time for each marking?

<p>
- In the results figures, what do the error bars show? Do they show quartiles, standard deviation, standard error, confidence intervals (for how much percent?), or something else?

---

### Decision · Program_Chairs · 2022-04-17

Reject